# Multi-Omics Alleviates the Limitations of Panel Sequencing for Cancer Drug Response Prediction

**DOI:** 10.3390/cancers14225604

**Published:** 2022-11-15

**Authors:** Artem Baranovskii, Irem B. Gündüz, Vedran Franke, Bora Uyar, Altuna Akalin

**Affiliations:** 1Non-Coding RNAs and Mechanisms of Cytoplasmic Gene Regulation Lab, Berlin Institute for Medical Systems Biology, Max Delbrück Center (MDC) for Molecular Medicine, Hannoversche Str. 28, 10115 Berlin, Germany; 2Integrative Cellular Biology & Bioinformatics Lab, Saarland University, 66123 Saarbrücken, Germany; 3Max Delbrück Center (MDC) for Molecular Medicine, Bioinformatics and Omics Data Science Platform, The Berlin Institute for Medical Systems Biology, Hannoversche Str. 28, 10115 Berlin, Germany

**Keywords:** multi-omics, cancer, drug response prediction, pharmacogenomics, panel sequencing

## Abstract

**Simple Summary:**

Cancer is a complex, heterogeneous collection of diseases with hundred of different subtypes. Genomic aberrations that are primarily thought to be the root causes of different cancers have been clinically used as evidence for both the diagnosis and also matching individual patients to proper treatment options. However, the complexity of cancer manifests itself differently in each patient when inspected at the molecular level. Even patients with the same cancer type rarely have identical root causes for the same disease. Without an extensive molecular profile of a patient, it has been challenging to match the patients to the best treatment options. To remedy this, comprehensive genomic profiling panels have been developed to monitor hundreds of genes for a given patient, which has helped broaden the treatment options for patients. However, genomic aberrations detected in such panels still do not reflect the full complexity of how a tumour responds to cancer drugs. In this study, we demonstrate that using an additional layer of molecular information (called the transcriptome) on top of genomic aberrations that can be detected with cancer gene panels can provide significant improvements in predicting the cancer drug response in pre-clinical cancer models. Thus, this study serves as a push towards incorporating the transcriptome measurements more routinely in (pre-)clinical practice.

**Abstract:**

Comprehensive genomic profiling using cancer gene panels has been shown to improve treatment options for a variety of cancer types. However, genomic aberrations detected via such gene panels do not necessarily serve as strong predictors of drug sensitivity. In this study, using pharmacogenomics datasets of cell lines, patient-derived xenografts, and ex vivo treated fresh tumor specimens, we demonstrate that utilizing the transcriptome on top of gene panel features substantially improves drug response prediction performance in cancer.

## 1. Introduction

Cancer is a collection of diseases characterized by abnormal cellular growth and the invasion of other body parts. It affected 19 million people in 2020 and was the cause of 9.5 million deaths that year alone [1]. Cancer has been primarily considered to be a disease of the genome, where the accumulation of alterations is the underlying cause of the transformation of normal cells into malignant cancerous cells with survival and proliferation advantages [2]. Genetic alterations of this kind have been studied to understand the mechanisms of cancer and to develop targeted therapies. The latter and companion diagnostic tools have transformed oncology [3], promising more precise treatments tailored to tumors’ genetic profiles. Various targeted therapies have been successfully developed to counteract the defects in the molecular machinery borne out of such oncogenic mutations [4,5,6]. To this date, most of the markers approved for targeted therapy decisions are single-gene markers [7]. It has thus become crucial to develop accurate, sensitive, and high-throughput genomic assays to accommodate the increasingly genotype-based therapeutic approaches. Commercial companies, such as Foundation Medicine, as well as large cancer research centers, such as Memorial Sloan Kettering and Dana Farber, have produced their panel sequencing assays to guide therapy for cancer patients [8]. These techniques examine genes that are frequently mutated in cancer to assess mutations and copy number variations. Especially for diagnostics, the approved methods for targeted drugs are usually the presence or absence of the mutations. Therefore, the assay developers focus primarily on mutation calling accuracy as a metric of the usefulness and accuracy of the assay [8].

Although comprehensive genomic profiling using cancer gene panels has demonstrated value in broadening the treatment options for patients based on matching a patient’s genomic lesions to cancer driver gene aberrations associated with FDA-approved treatment indications [8,9], the presence/absence of mutations in such genes does not necessarily translate into improved predictive power for estimating the patient’s response to the potential treatments. While for some drugs the variation in drug response can be explained by a very specific mutation (for instance, BRAF V600E mutation is a strong predictor for response to Vemurafenib in metastatic melanoma [4]), for many drugs the knowledge of the mechanism of action is missing. This is because many drugs are discovered via phenotypic screening of model systems rather than target-based approaches [10]. Of note, such single mutation markers for a given cancer type are not necessarily good markers for other cancer types. For instance, BRAF V600E, while a good predictor for metastatic melanoma, is a poor predictor of response in metastatic colorectal cancer [11]. More importantly, the latest compilation of the hallmarks of cancer recognized in the field includes factors such as the non-mutational epigenetic aberrations, the involvement of the immune system in the tumor microenvironment, and the composition of the microbiome along with genomic defects [12]. These layers of information cannot be sufficiently captured by focusing on the restricted set of genomic alterations and necessitate other data modalities. Among those, transcriptome profiling—besides being a cheap and accessible option in terms of logistics—has been shown to yield strong predictors of drug response [13,14,15].

Here, we set out to quantify the extent to which the usage of the transcriptome as an additional data modality improves the drug response prediction performance compared to only the genetic features restricted to the cancer gene panels (such as mutations and copy number variations). We leveraged publicly available pharmacogenomics datasets, including genomic and transcriptomic profiles and drug sensitivity measurements in three types of datasets: cancer cell lines (using the CCLE database [16] and PRISM project [17]), ex vivo treated fresh tumor specimens from Acute Myeloid Leukemia patients (BeatAML) [18], and patient-derived xenografts (PDX) [19]. These datasets span three vastly different model systems to anti-cancer drug efficacy testing, each of which varies in biological complexity and comes with unique challenges and advantages. Testing across these datasets should deliver an exhaustive assessment of the importance of transcriptomic features.

## 2. Results

In all three settings, with an application of out-of-the-box machine learning techniques (see Section 4), we modelled drug responses for all available drugs in two reported data modalities, using only panel gene features (panel (PS)) or using the transcriptomic features on top of panel features (multi-omics (MO)). While achieving only moderate predictive power (CCLE mean R-square ~10%, n = 396; BeatAML mean R-square ~12%, n = 106), the MO modalities of the CCLE and BeatAML datasets showed an overall increase in predictive power (up to a 5-fold improvement for certain drugs) in comparison to PS data (Figure 1A and Appendix A). Of note, we observed a significant positive correlation (*r* = 0.4, *p* < 0.0001) between the percentage of gene expression features among the top 100 features and an increase in MO’s predictive power over PS (Figure 1B). Modelling in xenografts generally conferred similar results. Ten out of twelve drugs showed a significant increase in MO’s predictive power of PS (Wilcoxon’s *p* < 0.05) (Figure 1C,D, Appendix A). These results were obtained by building random forest regression models; however, we have also reproduced similar findings, where transcriptome features added substantial predictive power on top of panel features, using both Elastic Net (GLMnet) and Support Vector Machines (with radial kernels) (Appendix A—see Section 4).

The improvement of MO over PS across all datasets is nearly univocal, yet heterogeneous. The most extreme improvement we have observed was for Venetoclax in beatAML dataset, where using the panel features yielded an R^2^ value of 0.03, and the MO features yielded an R^2^ value of 0.49. Hence, the top predictive features for Venetoclax consisted solely of cell type and cancer hallmark signatures (Appendix A). For some drugs, a 4- to 5-fold improvement in drug response prediction was recorded. For others, the improvement was modest (e.g., ~0.025 change in R^2^ between MO and PS). To an extent, this difference could be explained by a drug’s mechanism of action (MOA), as some perturb larger shares of cellular machinery than others. We selected the CCLE dataset as the most representative to test this (n drugs = 396) (Figure 1D and Appendix A). Among the drugs for which we observed the most improvement, there are histone deacetylase (HDAC) inhibitors, a relatively novel class of anti-cancer drugs that interferes with epigenetic regulation [20]. This MOA likely affects the transcriptome on a broad scale via secondary effects that arise from altered transcription. Likewise, topoisomerase inhibitors and bromodomain inhibitors drive wide transcriptional changes across the genome. The former inhibits the action of DNA topoisomerases that lead to the activation of the DNA damage response cascade [21]. Bromodomain inhibitors compete for the bromodomains of the respective proteins and prevent binding of the latter to acetylated histones and transcription factors [22]. Among the drugs with a defined target pathway, the effects of MO improvement are more modest. Altogether, the scale of off-target transcriptional perturbation seems to be beneficial for MO modelling, as it produces a signal outside of the defined panel’s reach.

## 3. Discussion

In this study, we set out to demonstrate two points. First, genomic features derived with comprehensive genomic profiling methods (panel sequencing) and used during clinical/pre-clinical drug development often have limited predictive power for drug response in cancer pre-clinical models. Second, we sought to elucidate how the drug response prediction could be improved using additional transcriptomic features. We showed that using cell type and cancer hallmark gene signature scores derived from the transcriptome on top of panel-derived features improves predictive power across different pre-clinical models (cell lines, patient-derived xenografts, ex vivo treated human samples), irrespective of the modelling method used. Our main aim in this study was not to argue that panel sequencing should be replaced by transcriptome profiling, but rather to demonstrate that using the transcriptomic features could have added benefit for treatment response prediction. The logic we follow is that if modelling drug responses using only panel features has limited power in pre-clinical models, then it would be even more limited for actual patients who would receive such treatments based on few marker genotypes, as the pre-clinical models cannot perfectly represent the complexity of the tumor or its microenvironment. Our second aim in this study was to look for general trends across drugs and datasets using off-the-shelf methods with a fair comparison of feature sets used in the modelling. We are mainly interested in the general trends; therefore, we refrain from making strong conclusions about individual drugs. However, we have quantified the predictive importance of all the studied features for each drug as Appendix A. It is important to note that the top markers we report would be correlative in nature. One would need to use causal inference methods to figure out drug-specific causal biomarkers.

Although the pre-clinical models we studied here do not perfectly reflect the complexity of the actual tumor microenvironment in a human patient, this kind of a large-scale data analysis of drug responses would not be feasible to carry out on actual patients due to logistical and ethical limitations. While replicating this kind of a large-scale analysis could not be extended to actual human samples, follow-up studies could address some questions that we have not addressed here. First of all, there could be many more alternatives in regard to how to preprocess the genomic and transcriptomic features in terms of converting the input data into less noisy and more information-dense latent features, optionally with added prior knowledge. For instance, mutation data could be converted into cancer mutational signatures, or transcriptome data could be integrated with prior knowledge networks to derive causal subnetwork features. Moreover, different layers of omics data modalities could be integrated using multi-omics integration methods. More sophisticated deep learning-based drug response modelling tools could be used, including pre-training and transfer learning approaches. Finally, the robustness and generalization power of the prediction models could be compared in a cross-dataset setting, where the models are built and validated in independently acquired resources or cross-tissue settings, where the models could be evaluated on cell lines derived from a tissue of interest unrelated to the tissues that are considered during the training procedure. However, our purpose here was not to benchmark the potential data processing or modelling algorithms, but rather to have a fair comparison of distinct feature sets in terms of their predictive power for drug response.

## 4. Methods

### 4.1. Data/Code Availability

In this study, the following publicly available pharmacogenomics datasets were used:

Cancer Cell Line Encyclopedia (CCLE) [16] downloaded from https://depmap.org/portal/download/, accessed on 11 April 2022. The drug response measurements from the PRISM project [17] were used for the corresponding CCLE samples.

Patient-Derived Xenografts (PDX) [19].

BeatAML: ex vivo drug sensitivity screening of acute myeloid leukemia patient tumor specimens [18] downloaded and processed using the PharmacoGx R package [23].

Annotation of drug classes was downloaded from Drug Repurposing Hub project [24].

All codes to download, process, analyze the datasets, and reproduce the figures in this manuscript can be found here: https://github.com/BIMSBbioinfo/multiomics_vs_panelseq, accessed on 11 April 2022.

### 4.2. Data Processing

Mutations: The mutation data were converted into a matrix of mutation counts per gene per sample. The resulting matrix was further filtered to only keep mutation data for genes in the OncoKB cancer gene list (https://www.oncokb.org/cancerGenes, accessed on 15 April 2022.). Mutation data was available in all three datasets.

Copy Number Variations: The copy number variation data was used as downloaded from the respective resources. Copy number variation data was also filtered to only keep genes found in the OncoKB cancer gene list. Both the CCLE and PDX datasets contained CNV data available, but it was not available for the BeatAML dataset.

Gene Expression (Transcriptome): To reduce the dimensionality and obtain less noisy features, the gene expression datasets were converted into gene-set activity scores using single-sample gene set scoring (singscore R package [25]). The gene sets utilized in this study were the Cancer Hallmarks gene signatures (50 gene sets) from the MSIGDB database [26] and tumor microenvironment-related gene sets (64 gene sets) curated in the xCell R package [27]. Gene expression data was available for all three datasets.

Drug Sensitivity Measures: For the CCLE and PDX datasets, AUC (area under the curve) scores derived from dose–response curves as published in the respective resources were used in the prediction models. For the BeatAML dataset, recomputed AAC (area above the curve) scores were used as downloaded via the PharmacoGx R package [23].

### 4.3. Drug Response Modelling with Machine Learning

For drug response modelling, we considered two main scenarios based on the availability of datasets. In the first set, we considered mutation and/or copy number variation data for genes found in the OncoKB cancer gene list, which aims to simulate panel-sequencing. In the second set, we considered the features as in the first set along with the whole transcriptome profiling as an additional data modality, which is further converted into gene-set activity scores. The second set represents the multi-omics condition, in which the panel features are concatenated with the transcriptome features (gene-set scores).

### 4.4. For Both Settings, All Three Datasets (CCLE, PDX, BeatAML) Were Analyzed with the Following Protocol

Only drugs that were treated on at least 100 samples were considered.

For each drug, the samples were split into training (70% of samples) and testing groups (30% of samples). See Appendix A for the specific sample counts used for each drug.

Caret R package [28] was used to build random forest regression models (using either ranger [29], logistic regression models (glmnet) [30], and support vector machines (svmRadial)) on the training data, where the genomic/transcriptomic features were used as predictors and the drug response values were used as the outcome variable. We used 3-times repeated 5-fold cross-validation for hyperparameter tuning to find the best model parameters based on the training data. Near-zero-variation filtering, scaling, and centering were applied as data processing steps. Applying principal component analysis (PCA) as a processing step led to poorer prediction results for both multi-omics and panel-seq features. However, the overall trend was the same, where MO-based models yielded better results than PS-based models (Appendix A); therefore, we excluded PCA processing when reporting the main results. For the PDX samples, this step was repeated 20 times by resampling the training/testing portions. This was only applied for the PDX samples, due to the small number of drugs (N = 12) treated on at least 100 samples.

The final model performance was evaluated on the testing data. Spearman rank correlation, R-square, and root mean squared error (RMSE) metrics were computed, and the R-square metric was used to report the results in the final figures.

Feature importance rankings were obtained using ‘caret::varImp’ function and are provided in Appendix A for each modelling method and for each drug.

Drugs were summarized according to their mechanism of action and were evaluated by mean multi-omics improvement.

## 5. Conclusions

We believe that in order to better understand cancer and develop better drugs and diagnostics, we need to make use of all the molecular features by integrating different omics datasets. In this manuscript, using multi-omics and machine learning techniques, we showed that multi-omics has indeed superior performance for drug response prediction in cancer.

## Figures and Tables

**Figure 1 cancers-14-05604-f001:**
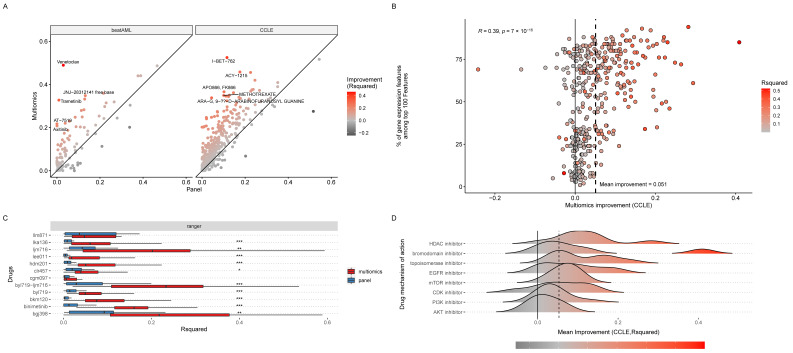
Evaluation of the performance of drug response prediction when using only panel-seq features (mutations and/or copy number variations) or using transcriptome features in combination with panel-seq features (multi-omics). (**A**) Improvement of multi-omics (as in R-square metric) in comparison to panel-seq features for the testing portion of BeatAML (**left** panel) and CCLE (**right** panel) datasets for 106 and 396 drugs, respectively. (**B**) Correlation of the prediction performance improvement (multi-omics vs. panel-seq) with respect to the proportion of transcriptome features among the top 100 most important predictors of drug response for CCLE datasets. (**C**) Improvement of multi-omics (in red) (as in R-square metric) in comparison to panel-seq features (in blue) for the testing portion of the PDX dataset for 12 drugs. Stars above the boxplots represent significance levels: *** for *p* < 0.001, ** for *p* < 0.01, * for *p* < 0.05. (**D**) Drug classes with loose pathway specificity show higher average improvement in MO over PS. Drug classes (*y*-axis) are ordered by average improvement in MO and filtered to keep only those that have a minimum of five drugs in a class. The dashed line corresponds to the global average improvement in MO (0.051) as reported in figure panel (**C**).

## Data Availability

All data analysed in this study have been previously published (See Section 4.1). The raw and processed data along with the code that was used to process, analyse, and visualise the findings reported in this study can be found at our GitHub repository: https://github.com/BIMSBbioinfo/multiomics_vs_panelseq (accessed on 11 April 2022).

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
