# Peer review of "Multi-Omics Alleviates the Limitations of Panel Sequencing for Cancer Drug Response Prediction"

_cancers, 2022, doi:10.3390/cancers14225604_

Round 1

Reviewer 1 Report (Previous Reviewer 3)

The authors answered my comment successfully. I recommend to accept the paper for publication in Cancers. Congratulations!

Author Response

We thank the reviewer for helpful feedback in previous round

Reviewer 2 Report (Previous Reviewer 2)

My comments:

1. In the results section on page 2, the authors claimed that  figure 1 b shows that "a significant positive correlation between the percentage of gene expression features among the top 100 features and an increase in MO's predictive power over PS".  I can see the increase in MO's predictive power over PS in figure 1 b. However, I couldn't see anything about the percentage of gene expression features. Can the authors explain figure 1b in a better way?

2. The authors claim that HDAC inhibitors' MOA is likely to affect the transcriptome on a broad scale and HDAC inhibitors are having the most improvement. Does it mean that MO doesn't improve prediction results for drugs with MOA affecting the transcriptome on much smaller scale?

Author Response

reviewer 2:

  1. In the results section on page 2, the authors claimed that  figure 1 b shows that "a significant positive correlation between the percentage of gene expression features among the top 100 features and an increase in MO's predictive power over PS".  I can see the increase in MO's predictive power over PS in figure 1 b. However, I couldn't see anything about the percentage of gene expression features. Can the authors explain figure 1b in a better way?

Response:

We thank the reviewer for this comment. In Figure 1, plots annotated as B and C were mislabelled resulting in confusion that was pointed out by the reviewer. To address this, we exchanged plots B and C in Figure 1 in accordance with the text in the legend and results section.

  1. The authors claim that HDAC inhibitors' MOA is likely to affect the transcriptome on a broad scale and HDAC inhibitors are having the most improvement. Does it mean that MO doesn't improve prediction results for drugs with MOA affecting the transcriptome on much smaller scale?

Response:

The reviewer brought up an interesting point. We detect an overall improvement in drug response prediction over P when using MO. In other words, MO improves predictions for drugs with a localized effect on transcriptome as well, however on a scale smaller than for the drugs with a broad transcriptome perturbation effects (HDAC, bromodomain inhibitors, topoisomerase inhibitors)(Figure 1D). . The latter drug types produce more transcriptomic perturbations and thus generate more potential signal that goes undetected by features selected for sequencing panels, while the same signal can be harvested and utilised from a detailed transcriptomic snapshot provided by multi-omics.

This manuscript is a resubmission of an earlier submission. The following is a list of the peer review reports and author responses from that submission.

Round 1

Reviewer 1 Report

Heterogeneity between individual patients are challenges for cancer patient clinic management. Sensitive predictive biomarker(s) is crucial for the success of precision medicine. This is an interesting short paper with a great use of publicly available pharmacogenomics datasets for better predict of anticancer drug responses from cancer cell lines, PDX models, as well as ex vivo treated fresh tumor specimens. The authors showed that multi-omics has superior performance than cancer gene panels for drug response prediction in cancer. However, there are some concerns about this manuscript:

This manuscript is only subdivided into a "Main" and "methods" and the authors seem to discuss their results prior to explaining methodology.

The authors use the abbreviation PCA in Line 154 but never explain what PCA is – is it for “Principal component analysis”? It needs to be specified.

How would this work translate to assisting clinical practice, when dealing with a specific type of cancer?

The authors also did not discuss the potential weaknesses/biases in their own study, which would be helpful to contextualize their methods for the reader.

Author Response

"This manuscript is only subdivided into a "Main" and "methods" and the authors seem to discuss their results prior to explaining methodology."

Response: We have introduced the necessary section headers (Abstract/Introduction/Results/Methods) to fit the structure of the journal’s requirements. 

"The authors use the abbreviation PCA in Line 154 but never explain what PCA is – is it for “Principal component analysis”? It needs to be specified."

Response: It has been clarified in the methods section that we mean “Principal Component Analysis” before the first usage of the abbreviation.

"How would this work translate to assisting clinical practice, when dealing with a specific type of cancer?

The authors also did not discuss the potential weaknesses/biases in their own study, which would be helpful to contextualize their methods for the reader."

Response: We thank the referee for pointing this out. We agree that there should have been a more detailed discussion section. We added a more detailed “Discussion” section where we discuss the implications, weaknesses, and possible improvements that could be made in follow-up studies of this work. 

Reviewer 2 Report

Drug response prediction is very challenging and crucial in cancer research. With advanced NGS technologies, more and more multi-omics data become available for better drug response prediction. Many researchers have utilised multi-omics data to predict responses of singe drugs or combinations in the past few years.

My major concern: the conclusion was based on using a single model: random forest regression. Would other models such as linear regression and xgboost give the same conclusion? 

Author Response

"Drug response prediction is very challenging and crucial in cancer research. With advanced NGS technologies, more and more multi-omics data become available for better drug response prediction. Many researchers have utilised multi-omics data to predict responses of singe drugs or combinations in the past few years.

My major concern: the conclusion was based on using a single model: random forest regression. Would other models such as linear regression and xgboost give the same conclusion? "

Response: We thank the referee for the suggestion. In the initial submission, we had used Elastic Nets (GLMnet) for CCLE and PDX samples and Random Forests for the beatAML dataset. However, we followed the suggestion of the referee and re-built all the models using three different methods: random forests, elastic nets, and support vector machines with “radial” kernels. We had issues installing the “xgboost” library and due to the time constraint for the revision, we used support vector machines instead. In the main figures we report the results we obtained using random forests for all datasets, because we got the best performance for both panel features and for the multi-omics features using random forests. However, both elastic nets and support vector machines yielded similar results, in other words, we could reproduce our finding that using the transcriptome features on top of the panel features leads to substantial improvements in predictive power for drug response prediction. We report the main findings based on random forests, but we also provide all findings using elastic nets and support vector machines in supplementary figures and supplementary tables.

Reviewer 3 Report

In this manuscript Baranovskii et al. demonstrate that drug response prediction in cancer based on gene panel features can be improved by adding data from transcriptomics and machine learning. The rationale and methodology behind the study are scientifically sound. The results indicate an improvement in the predictive power of the multiomic approach, which is not surprising and similar approaches have been published previously (e.g. https://doi.org/10.1093/bioinformatics/btz318 ). In order to capitalize on the data generated on the authors and add novelty to the study, did the authors find any drug classes or tumor types in which the RNAseq data were especially beneficial as a predictor? The venetoclax as an outlier is quite interesting. Would it be possible to provide the potential reader with additional data regarding which gene sets are particularly important for the improved prediction in the case of venetoclax? We still do not quite know what causes venetoclax resistance/sensitivity and, to make matters worse, it seems to be cancer-specific.

Author Response

"In this manuscript Baranovskii et al. demonstrate that drug response prediction in cancer based on gene panel features can be improved by adding data from transcriptomics and machine learning. The rationale and methodology behind the study are scientifically sound. The results indicate an improvement in the predictive power of the multiomic approach, which is not surprising and similar approaches have been published previously (e.g. https://doi.org/10.1093/bioinformatics/btz318 ). In order to capitalize on the data generated on the authors and add novelty to the study, did the authors find any drug classes or tumor types in which the RNAseq data were especially beneficial as a predictor? The venetoclax as an outlier is quite interesting. Would it be possible to provide the potential reader with additional data regarding which gene sets are particularly important for the improved prediction in the case of venetoclax? We still do not quite know what causes venetoclax resistance/sensitivity and, to make matters worse, it seems to be cancer-specific."

Response: We thank the referee for the suggestions. We have re-built all drug response models using random forests, elastic nets, and support vector machines. While doing so, we have also compiled the feature importance metrics for all models and drugs. Now, we provide feature importance metrics as supplementary tables. We have also added a new figure looking into the rankings of drug classes based on the benefit of using the transcriptomics features on top of panel features. We added a new paragraph describing this result. Briefly, we find that HDAC inhibitors, Topoisomerase inhibitors, and Bromodomain inhibitors, which are known to affect general transcriptional regulation, seem to benefit the most from including the transcriptome in the prediction models, while drugs that have more direct targets seem to benefit less from this exercise. These results are reproduced using all three methods (random forests, elastic nets, support vector machines). Moreover, we provide a feature ranking for Venetoclax as a supplementary figure (Suppl. Figure 2c). In fact, all of the top 20 features actually consist of cell-type or cancer-hallmark gene-set signature scores derived from the transcriptome. We would also like to clarify that the purpose of this study was to observe the general trends across many drugs rather than studying individual drugs’ response signatures, so we refrain from making strong conclusions for individual drugs. Also the top features we derive from this exercise will be correlative in nature, as we did not carry out a causal-inference procedure, which could be done also by incorporating prior knowledge networks. However, for this small communication paper, this would be beyond the scope of our purposes.